# Quality assurance practices in tuberculosis diagnostic health facilities in Ethiopia

**Yeshiwork Abebaw**[1,2]*, **Abebaw Kebede**[1], **Kirubel Eshetu**[1], **Ephrem Tesfaye**[1], **Mengistu Tadesse**[1], **Waganeh Sinshaw**[1,2], **Misiker Amare**[1], **Dinka Fikadu Gamtesa**[1], **Betselot Zerihun**[1,2], **Melak Getu**[1,2], **Getachew Seid**[1], **Anteneh Yalew**[1,3,4], **Getu Diriba**[1]

**1** Ethiopian Public Health Institute, Addis Ababa, Ethiopia, **2** Department of Microbiology, Immunology, and Parasitology College of Health Sciences, Addis Ababa University, Addis Ababa, Ethiopia, **3** Division of Epidemiology and Biostatistics, Department of Global Health, Faculty of Medicine and Health Sciences, Stellenbosch University, Cape Town, South Africa, **4** Department of Statistics, College of Natural and Computational Sciences, Addis Ababa University, Addis Ababa, Ethiopia

* Yeshi885@gmail.com

## Abstract

### Introduction

The quality of tuberculosis laboratory services in health facilities is a mandatory component of detecting active pulmonary TB cases and treatment follow-up. However, ensuring the quality of laboratory test results is a concern. This study aimed to assess the quality assurance practices in the tuberculosis diagnostic health facilities of Ethiopia.

### Materials and methods

A cross-sectional study was conducted from October 2018 to March 2019 at nine governmental TB-culture laboratories and 34 randomly selected GeneXpert® MTB/RIF (Xpert® MTB/RIF) testing health facilities in Ethiopia. Participating health facilities were interviewed and laboratory documents and records present since 2017 were observed. Prior to the data collection, training was given to the data collectors. Descriptive statistics were used to produce results and were presented with tables and graphs.

### Results

From a total of 34 Xpert® MTB/RIF testing laboratories, 50% run Internal Quality Control (IQC) for Acid-Fast Bacillus (AFB) Microscopy and 67.6% had lot-to-lot verification of staining reagents. For the Xpert® MTB/RIF assay, a lot-to-lot verification of cartridge and method validation was performed only in 8.8%and 20.6% of Xpert® MTB/RIF testing laboratories respectively. All TB-culture laboratories included in the study ran negative control (start and end IQC) during TB-culture sample processing and performed lot-to-lot verification for Mycobacteria Growth Indicator Tube (MGIT) in 88.9% of TB-culture laboratories. External Quality Assessment (EQA) Proficiency Testing (PT) for AFB microscopy is practiced in 79.4% Xpert® MTB/RIF testing laboratories and 100.0% for the Xpert® MTB/RIF assay. TB-Culture PT participation practice among TB-culture laboratories was 88.9%. A major

**Competing interests:** The authors have declared that no competing interests exist.

**Abbreviations:** AFB, Acid-fast bacilli; DST, Drug susceptibility testing; EQA, External quality assessment; HC, Health Center; EPHI, Ethiopian Public Health Institute; LPA, Line Probe Assay; LJ, Lowenstein Jensen; SLMTA, Strengthening Laboratory Management Toward Accreditation; MGIT, Mycobacterial growth indicator tubes; NTRL, National TB reference laboratory; QA, Quality assurance; QAS, Quality assurance System; QC, Quality Control; QMS, Quality management systems; SOP, Standard operating procedure; TB, Tuberculosis; PT, Proficiency testing; WHO, World Health Organization.

challenge for health facilities during PT participation was the AFB PT-sample transportation delay (40.7%) and the Xpert® MTB/RIF assay EQA-PT feedback missing (38.2%).

## Conclusion

This assessment reveals that IQC for AFB microscopy, lot-to-lot verification, method validation, and equipment calibration were not well-practiced. The majority of TB diagnostic health facility laboratories had EQA-PT participation practice, but a significant gap in PT-sample transportation and missing feedback was identified.

## Introduction

Tuberculosis (TB) laboratory plays an essential role in diagnosing TB and monitoring its treatment [1]. In the last decade, TB screening methods have been improved beyond smear microscopy, TB culture, and drug susceptibility testing (DST) using solid media [1]. Innovations in TB diagnostics (near point-of-care testing) have ensured accessibility and affordability in low and middle-income countries [1, 2]. In addition to the expansion of TB testing, ensuring the quality assurance of laboratory services is an important element for facilitating TB controlling efforts [3].

Quality Assurance (QA) is designed to continuously improve the reliability and efficiency of laboratory services in order to achieve the required technical quality. It consists of Quality Control (QC), External Quality Assessment (EQA), and continuous quality improvement [3, 4]. Quality control is intended to monitor routine technical laboratory activities to control errors in the performance of tests [5, 6]. EQA is also a valuable tool for assessing technology in use and identifying gaps in laboratory performance [7, 8]. It allows participating laboratories to assess their capabilities by comparing their results with those in other laboratories in the network [7, 9]. With this, the World Health Organization target enrolling all TB diagnostic facilities with EQA in 2020 [10]. EQA for tuberculosis diagnosis and drug resistance is practiced in different countries with proficiency testing and subsequent technical assistance to improve the DST quality of participating laboratories [11, 12].

Strengthening Laboratory Management towards Accreditation (SLMTA) is a competency-based management training program designed to bring immediate and measurable improvement in laboratory quality and services in resource-limited settings [13]. Ethiopia joined the SLMTA program for the first time in 2010, using selected health facilities [14]. Later, TB-SLMTA was introduced in TB culture and DST laboratories in 2013. Since then, the Ethiopia National TB Prevention and Control program with partners has been working to expand the customized quality management system all over the country's TB diagnostic facilities [15]. Subsequently, regional health institutes or sub-regional laboratories, or designated EQA centers have been conducting onsite supervision and blinded rechecking of TB slides in their respective catchment areas based on national EQA guidelines [16]. The National TB Reference Laboratory (NTRL) of Ethiopian Public Health Institute (EPHI) provides PT for the Xpert® MTB/RIF assay, TB culture and identification, and Line Probe Assay (LPA) first and second-line DST for TB diagnostic facilities. Additionally, One-World Accuracy, an international PT provider, supplies PT for selected TB diagnostic health facilities. Also, CDC-Atlanta offers PT for selected Xpert® MTB/RIF testing sites.

There are national and international recommendations for how often QC and EQA should be performed for each TB testing method [17–20]. However, many health facilities are

challenged in the implementation of QC and participation in EQA because of several factors such as poor infrastructure, low human resource capacity, and inappropriate technologies [19, 20]. In Ethiopia, some studies have been conducted evaluating the QA practices of TB diagnostic centers [8, 21]. However, these studies did not cover TB culture and Xpert® MTB/RIF testing laboratories, while the country currently recommends the Xpert® MTB/RIF assay for rapid detection of TB and TB culture for MDR-TB treatment monitoring. Therefore, this study aimed to assess the QA practices of TB culture and Xpert® MTB/RIF testing laboratories in Ethiopia.

## Materials and methods

### Study design and setting

A cross-sectional study was conducted in 31 selected governmental Xpert® MTB/RIF testing laboratories and nine TB culture laboratories from October 2018 to March 2019. Ethiopia is the second-most populous nation in Africa, with an estimated 112.0 million people in 2019 [22]. In the country, there are ten TB and DST culture laboratories. During the study period, there were 170 Xpert® MTB/RIF testing health facilities, while recently the machines were distributed to more than 300 TB diagnostic health facilities. The NTRL of EPHI is a national reference center and has been the sole PT provider for molecular testing (Xpert® MTB/RIF assay, Genotype MTBDRplus assay, and Genotype MTBDRsl assay) and PT for TB culture in Ethiopia.

### Sampling techniques and sample size

Stratified random sampling was used. The sample size was determined for facility-based assessment surveys by computing the following formula [23]:

$$n = \dot{\iota}$$

Where n is the sample size to be determined, the value of Z for 95% power is 1.96, p = 50%, and d is the margin of allowable error (0.15). Therefore, based on the above formula, all (nine) governmental TB-culture laboratories for TB-Culture and LPA quality assurance assessment and 34 Xpert® MTB/RIF testing health facilities for GeneXpert® MTB/RIF assay and Microscopy tests quality assurance assessment were included in the study.

During random selection of Xpert® MTB/RIF testing health facilities, three TB culture laboratories that provide Xpert® MTB/RIF tests and thirty-one health facilities that provide only GeneXpert® MTB/RIF assay and Microscopy tests were included. Therefore a total of 40 health facilities were included in the study with six health facilities for TB-culture and LPA only, three health facilities for TB-culture, LPA, GeneXpert® MTB/RIF assay, and Microscopy testing, and thirty-one health facilities for GeneXpert® MTB/RIF assay and Microscopy.

## Inclusion criteria

All governmental TB-culture and Xpert® MTB/RIF assay testing health facilities with at least one year of healthcare experience in the area were included in the study.

### Data collection methods

The study assessed the main QA practices in selected TB diagnostic centers for TB-culture and identification, LPA, Xpert® MTB/RIF assay, and AFB-Microscopy testing. The assessment covered internal quality control, a lot to lot verification, method verification, equipment maintenance, and EQA activities. We used a standardized checklist depending on the testing

services in the diagnostic centers. The questionnaire was created with the scope of service provided by TB diagnostic laboratories in mind (for TB culture laboratories and Xpert® MTB/RIF testing health facility laboratories). The laboratory head /quality officer of participating health facilities was interviewed. Depending on the question, we also observed laboratory documents and records present since 2017 to cross-check their QA practices. If the document or records were not present, it is considered not practiced. Prior to the data collection, training was given to the data collectors at NTRL on the data collection tools and then on-site evaluations were conducted.

## Data management and analysis

A code number was assigned to each health facility participating in the study to ensure confidentiality. Data were entered and analyzed using SPSS version 25 (SPSS Inc., Chicago, Illinois, USA). The frequency and percentage of each variable were calculated. The table and figures were used for the presentation of the data.

## Ethical clearance

The study was first approved by the Institutional Review Board (IRB) of the Ethiopian Public Health Institute with the reference number EPHI-IRB-120-2018. Written informed consent was collected from all participating health facilities before the interview. All study subjects have been informed of the purpose of the study and their right to anonymity. All the information obtained from the study subjects was coded to maintain confidentiality.

## Results

### General information on the testing laboratories' service capacity

A total of 40 health facilities participated in this study, including nine governmental TB culture laboratories and 31GeneXpert® MTB/RIF assay and Microscopy testing laboratories. From nine TB-culture laboratories, three TB culture laboratories were also assessed for quality assurance practice of GeneXpert® MTB/RIF assay and AFB microscopy. Among the selected laboratories, 27/34 (79.4%) Xpert® MTB/RIF testing laboratories and 7/9 (77.8%) TB-culture laboratories were enrolled in the SLMTA program. However, among health facilities enrolled in the SLMTA program, only 1/27 (3.7%) were accredited by the Ethiopian National Accreditation Office (ENAO) for AFB-microscopy and GeneXpert MTB/RIF assay, and 1/7 (14.2%) for TB culture and identification (ID) as indicated in Table 1.

### Sample collection and transportation system

In this study, more than half of Xpert® MTB/RIF testing laboratories 19/34 (55.9%), and TB-culture laboratories 6/9 (66.7%) had no tools for monitoring of delayance of samples after collection when receiving samples. Four (11.8%) Xpert® MTB/RIF testing health facilities received samples from health facilities that did not use standard triple packaging. Three (33.3%) TB-culture laboratories and 24 (70.5%) Xpert® MTB/RIF testing laboratories did not calibrate their thermometers regularly as shown in Table 1.

### Availability of standard operating procedure

Of all Xpert® MTB/RIF testing laboratories, lot-to-lot verification SOP was available in less than 25% of the laboratories (17.6%) for both AFB-microscopy and Xpert® MTB/RIF assay. Also, SOP for Method verification of Xpert® MTB/RIF assay was available only 18/34(52.9%) Xpert® MTB/RIF testing laboratories. In TB-culture laboratories, method verification and

**Table 1. General information on quality assurance practice in TB culture laboratories and Xpert testing health facilities.**

| | | | Gene Xpert® MTB/RIF assay testing health facilities n (%) | TB culture laboratories n (%) |
|---|---|---|---|---|
| Study sites | | Health Center | 8/34(23.5%) | 0 |
| | | Hospital | 24/34(70.5%) | 2/9(22%) |
| | | Institutional laboratory | 2/34(6%) | 7/9(78%) |
| Number of health facilities enrolled in the SLMTA program | | Yes | 27/34 (79.4) | 7/9 (77.8) |
| | | No | 7/34 (20.6) | 2/9 (22.2) |
| **Accredited test among health facilities enrolled in the SLMTA program** | | | | |
| AFB Microscopy | | Yes | 1/27 (3.7) | NA |
| | | No | 26/27(96.3) | NA |
| GeneXpert MTB/RIF assay | | Yes | 1/27(3.7) | NA |
| | | No | 26/27(96.3) | NA |
| Culture and ID | | Yes | NA | 1/7(14.2) |
| | | No | NA | 6/7(85.7) |
| LPA | LPA1st line | Yes | NA | 0 |
| | | No | NA | 7(100) |
| | LPA 2nd line | Yes | NA | 0 |
| | | No | NA | 7(100) |
| **Laboratory activities** | | | | |
| Monitoring of delayance after sample collection during the reception | | Yes | 19(55.9) | 6(66.7) |
| | | No | 15(44.1) | 3(33.3) |
| Ensuring sample transportation box kept cool during transport | | Yes | 19(55.9) | 6(66.7) |
| | | No | 15(44.1) | 3(33.3) |
| Way of specimens transportation | Mailing or courier system | Yes | 25(73.5) | 7(77.8) |
| | | No | 9(26.5) | 2(22.2) |
| | Hand carrying by staff | Yes | 10(29.4) | 5(55.6) |
| | | No | 24(70.6) | 4(44.4) |
| | Hand carrying by patient | Yes | 4(11.8) | 5(55.6) |
| | | No | 30(88.2) | 4(44.4) |
| Use of standard triple packaging | | Yes | 30(88.2) | 9(100) |
| | | No | 4(11.8) | 0 |
| Rejection logbook and rejection criteria | | Yes | 21(61.8) | 8(89.9) |
| | | No | 13(38.2) | 1(11.1) |
| Separate refrigerator for sample storage | | Yes | 19(55.9) | 8(89.9) |
| | | No | 15(44.1) | 1(11.1) |
| Routine preventive maintenance of refrigerators | | Yes | 23(67.6) | 6(66.7) |
| | | No | 11(32.4) | 3(33.3) |
| Refrigerator temperature monitoring | | Yes | 23(67.6) | 8(89.9) |
| | | No | 11(32.4) | 1(11.1) |
| Thermometer calibration | | Yes | 10(29.4) | 6(66.7) |
| | | No | 24(70.6) | 3(33.3) |

LPA: Line probe Assay; ID: Identification; TB: Tuberculosis; AFB: Acid fast bacillus; QMS: Quality management system; RIF: Rifampicin; n: Number of participants; NA: None applicable; MTB: *Mycobacterium Tuberculosis*; SLMTA: Strengthening Laboratory Management toward Accreditation; TB: Tuberculosis.

**Table 2. SOP available for TB diagnostic tests among 34 Xpert testing health facilities and nine TB-culture laboratories in Ethiopia.**

| Selected SOP and logbooks | | 34 GeneXpert® MTB/RIF assay testing health facilities | | | Nine TB-culture laboratories | | |
|---|---|---|---|---|---|---|---|
| | | Microscopy n (%) | | GeneXpert MTB/RIF assay n (%) | Culture and ID n (%) | | LPA n (%) |
| | | ZN | FM | | Processed Samples | Prepared Media | |
| SOP for Internal Quality Control | Yes | 24(70.6) | 23(67.6) | 31(91.2) | 7(77.8) | 8(88.9) | 6(66.7) |
| | No | 7(20.6) | 6(17.6) | 3(8.8) | 2(22.2) | 1(11.1) | 3(33.3) |
| | NA | 3(8.8) | 5(14.7) | N/A | N/A | N/A | N/A |
| Method Verification | Yes | NA | | 18(52.9) | 7(77.8) | | 6(66.7) |
| | No | NA | | 16(47.1) | 2(22.2) | | 3(33.3) |
| Lot Verification for new reagents | Yes | 6(17.6) | | 6 (17.6) | 5(55.6) | | 5(55.6) |
| | No | 28(82.4) | | 28(82.4) | 4(44.4) | | 4(44.4) |
| Continuous Quality Improvement | Yes | 22(64.7) | | 22(64.7) | 7(77.8) | | 7(77.8) |
| | No | 12(35.3) | | 12(35.3) | 2(22.2) | | 2(22.2) |
| Equipment Service Maintenance | Yes | 20(58.8) | | 22(64.7) | 7(77.8) | | 7(77.8) |
| | No | 14(41.2) | | 12(35.3) | 2(22.2) | | 2(22.2) |
| EQA Guideline/SOP | Yes | 21(61.8) | | 18(52.9) | 6(66.7) | | 6(66.7) |
| | No | 13(38.2) | | 16(47.1) | 3(33.3) | | 3(33.3) |
| Standard Request And Reporting Form | Yes | 30(88.2) | | 30(88.2) | 9(100) | | 9(100) |
| | No | 4(11.8) | | 4(11.8) | 0 | | 0 |
| Standard Registration Log Book | Yes | 31(91.2) | | 27(79.4) | 9(100) | | 9(100) |
| | No | 3(8.8) | | 7(20.6) | 0 | | 0 |

LPA:Line probe Assay; ID: Identification; TB: Tuberculosis; RIF: Rifampicin; n: Number of the participant; MTB: *Mycobacterium Tuberculosis*; ZN: Ziehl-Neelsen; FM: Fluorescence microscopy; NA: Not Applicable.

lot-to-lot verification SOP for TB-culture and Identification were 7/9(77.8%) and 5/9(55.6%) respectively as illustrated in Table 2.

## Quality assurance practice for AFB Smear microscopy testing and GeneXpert® MTB/RIF assay

Batch-to-batch testing of AFB reagents was performed in 67.6% of the laboratories, and AFB-IQC was run in 50% of Xpert® MTB/RIF testing laboratories (Fig 1).

Regarding quality assurance practices of GeneXpert® MTB/RIF assay, only 29.4% of Xpert® MTB/RIF testing laboratories monitored the Probe Check Control (PCC) and Specimen Processing (SPCC) (Fig 2). A lot-to-lot verification of new cartridges was performed in 3/34 (8.8%) of the laboratories. Seven (20.6%) Xpert® MTB/RIF testing laboratories have performed the method verification. Annual GeneXpert machine calibration was performed in 20/34 (58.8%) Xpert® MTB/RIF testing laboratories. However, 11/34 (32.4%) of them ran the samples with a machine that was not calibrated.

## Quality assurance practice for TB-culture and identification, and LPA based drug susceptibility testing

All TB-culture laboratories apply both MIGT and LJ media for TB-culture processing. During sample processing, a negative control was included with each batch and 3/9 (33.3%) of them included a positive control periodically. Internal quality control for the lateral flow assay (MPT64 Ag testing kit) was performed in 5/9 (55.6%) of the TB-culture laboratories. In this study, the number of TB-culture laboratories that verify lot-to-lot or batch-to-batch reagents ranged from 8/9 (88.9%) for MGIT media, growth supplement, and PANTA to 3/9 (33.3%) for

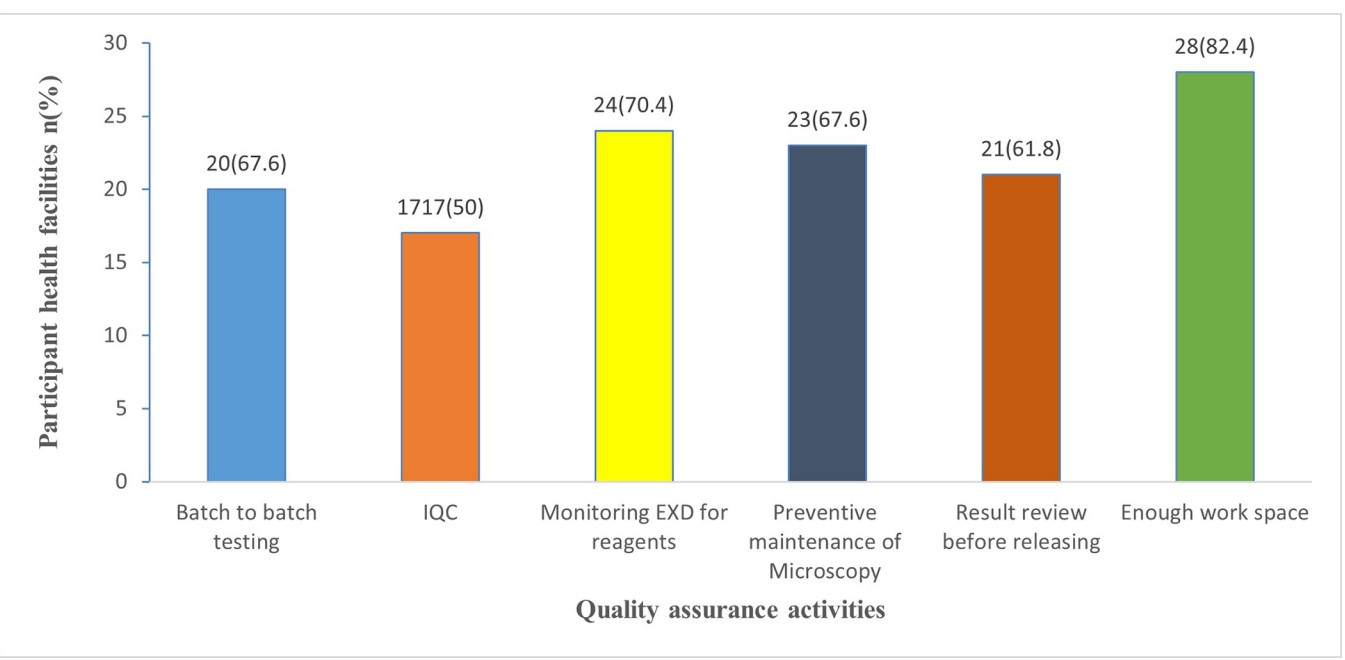

**Fig 1. Practice of quality assurance activities for microscopy.** EXD: Expiration Date; IQC: Internal Quality Control; n: Number of participant health facilities.

the lateral flow assay. Method verification for the MGIT machine and MPT64 Ag testing was performed only in 3/9 (33.3%) TB-culture laboratories. Also, almost 50% of TB-culture laboratories 4/9 (44.4%) did not have regular negative pressure calibration.

Internal quality control for LPA first and/or second-line were performed in all nine TB-culture laboratories including positive and negative controls with each batch of specimens/

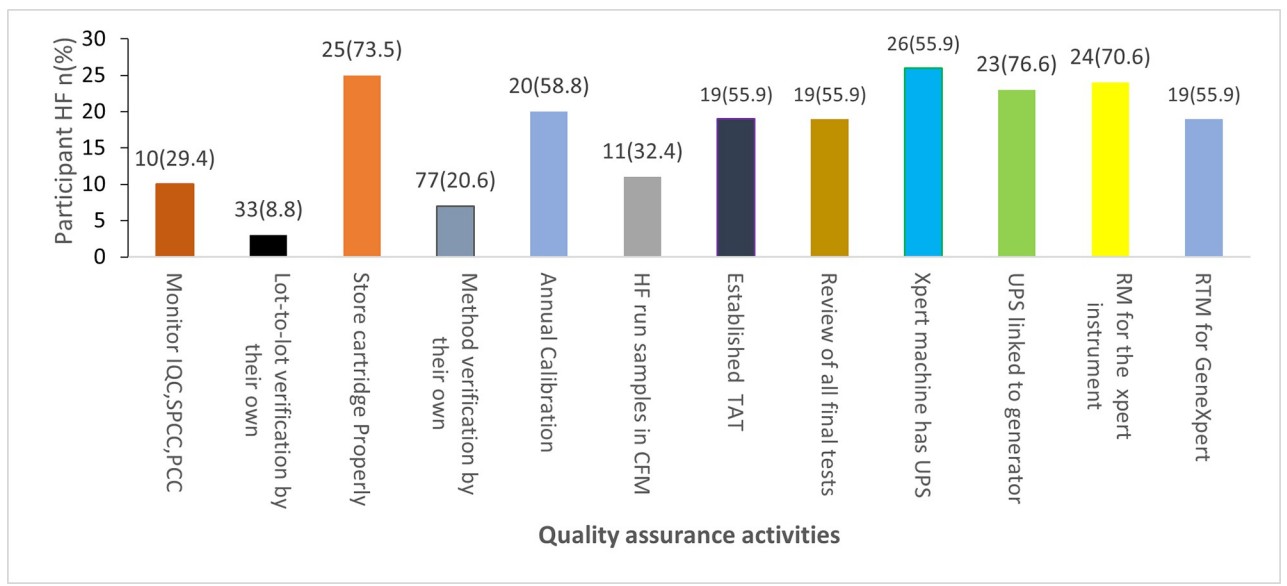

**Fig 2. Practice of quality assurance activities for Gene Xpert® MTB/RIF assay.** IQC: Internal Quality Control; PCC: Probe Check Control; SPCC: Specimen Processing; HF: Health Facility; CFM: calibration failed Module; UPS: Uninterruptible power supply; TAT: Turnaround time; RTM: Room temperature monitoring; RM: Routine maintenance; n: Number of health facilities.

cultures or one batch per week when LPA first and/or second-line performed. New lot verification for LPA was performed in only 3/9 (33.3%) of TB-culture laboratories and 5/9 (55.6%) of them monitored the expiration date and storage condition of kits (Table 3).

### External quality assurance practice in TB-diagnostic laboratories

Participating in EQA-blinded rechecking was observed in 18/34 (52.9%) Xpert® MTB/RIF testing health facilities for AFB-microscopy where the majority of them (18/39%) participate quarterly. Also, PT-participation practice for AFB-microscopy and Xpert® MTB/RIF assay were 27/34 (79.4%) and 34 (100%) respectively. However, this study observed that there was no similar PT participation program. Among the participants, 19/34 (55.9) participated bi-annually in the Xpert® MTB/RIF assay. Among TB-culture laboratories, PT-participant for TB-Culture and ID were 8/9 (88.9%) and 9 (100%) for LPA. On-site supervision practice was not well practiced in TB-culture laboratories, only 3/9(33.3%) by comparison with the Xpert® MTB/RIF assay, which was 30/34 (88.2%) (Table 4).

### Challenges of external quality assurance practice

This study identified 11/27 (40.7%), 8/27 (29.6%), and 9/27 (33.3%) Xpert® MTB/RIF testing laboratories that had challenges with delayed PT-sample transportation, communication gaps with the provider, and feedback missing respectively during AFB PT-participation. Major gaps identified during Xpert® MTB/RIF assay PT-participation were feedback missing 13/34 (38.2%) and delays in feedback from provider 12/34 (35.3%). The problem was a relative decline in TB-Culture Laboratories PT participation, as indicated in Table 4.

## Discussion

Implementation of quality assurance practice in TB laboratories plays a critical role in meeting the 2035 end-TB strategic goals [24]. In our study, 77.8% TB-Culture laboratories and 79.4% Xpert® MTB/RIF testing laboratories were enrolled in the SLMTA program. According to the literature, the implementation of the SLMTA program resulted in the provision of quality-assured laboratory services for patient care [25] and assisted health facilities' laboratories to reach test accreditation [26–33].

Among health facilities enrolled in the SLMTA program, only 3.7% of Xpert® MTB/RIF testing laboratories were accredited by the ENAO based on ISO 18519 standards for GeneXpert MTB/RIF assay and AFB-Microscopic test. Similarly, only 1/7 (14.2%) of TB-Culture laboratories have achieved accreditation for TB-Culture and ID testing. ENAO was established in Ethiopia in 2011 to supply accreditation services and became a full member of the International Laboratory Accreditation Cooperation (ILAC) in 2017 [34]. Accreditation of laboratories based on the ISO15189 standards increases the functioning of laboratories in terms of monitoring and improving the entire testing process [35].

The Xpert® MTB/RIF machine has been used as the primary TB diagnostic test in Ethiopia. Most Xpert® MTB/RIF testing laboratories serve as referral sites since the machines are not distributed to all health facilities [36]. We observed that one-quarter of TB-culture laboratories and nearly half of Xpert® MTB/RIF testing health facilities did not monitor the delayance of samples after collection when receiving samples which become difficult in tracing early problems before processing. Studies have shown that the most frequent sources of errors have occurred in laboratories that are in the pre-analytical phases; these account for 48%-62% of the total errors [37].

According to this study, 11.8% of Xpert® MTB/RIF testing health facilities received samples from health facilities that did not use standard triple packaging, which violated local and

**Table 3. TB-culture laboratories those practiced quality assurances activates for TB-culture and identification among nine TB-culture laboratories of Ethiopia.**

| Activities | | Frequency and percent | |
|---|---|---|---|
| | | Yes, n (%) | No, n(%) |
| Regular BSC calibration | | 9(100) | 0 |
| Regular negative pressure calibration | | 5(55.6) | 4(44.4) |
| Include end and start control (negative control) | | 9(100) | 0 |
| Include a positive control | | 3(33.3) | 6(66.7) |
| Perform lot to lot verification for media(MGIT tube), Growth Supplement, and PANTA | | 8(88.9) | 1(11.1) |
| Monitor the storage conditions and expiration date of the lyophilized PANTA | | 5(55.6) | 4(44.4) |
| Monitor expired date of reconstituted PANTA | | 7(77.8) | 2(22.2) |
| Proper storage and monitoring of the expired date of MGIT reagent | | 4(44.4) | 5(55.6) |
| Proper storage and monitoring of the expired date in-house reagent and media | | 7(77.8) | 2(22.2) |
| Testing each new lot/batch of in-house prepared media for sterility, growth performance | NALC-NaOH, | 5(55.6) | 4(44.4) |
| | 4%NaOH | 5(55.6) | 4(44.4) |
| | BHI/blood agar | 4(44.4) | 5(55.6) |
| | PBS buffer *(including PH)* | 5(55.6) | 4(44.4) |
| | LJ media | 5(55.6) | 4(44.4) |
| MGIT instrument preventive maintenance. | | 7(77.8) | 2(22.2) |
| Routine preventive maintenance for incubator | | 6(66.7) | 3(33.3) |
| Thermometry calibration for incubators | | 6(66.7) | 3(33.3) |
| Annual MGIT machine calibration | | 7(77.8) | 2(22.2) |
| MGIT machine method verified by your laboratory | | 3(33.3) | 6(66.7) |
| Recording growth results in weekly form LJ | | 9(100) | 0 |
| Preparing a blood agar plate for all positive cultures | | 8(88.9) | 1(11.1) |
| Inspect all negative cultures for growth prior to discarding. | | 8(88.9) | 1(11.1) |
| Incubate negative result for a negative result for 8 weeks | | 9(100) | 0 |
| Confirm the presence of MTB vs NTM using a Lateral flow assay | | 9(100) | 0 |
| Including positive and negative controls periodically for Lateral flow assay | | 5(55.6) | 4(44.4) |
| Proper storage and monitoring of the expired kit | | 7(77.8) | 2(22.2) |
| Lateral flow assay method verified by your laboratory | | 3(33.3) | 6(66.7) |
| Re-incubating and retesting all BHI negative and Smear negative | | 7(77.8) | 2(22.2) |
| Testing new lot numbers of Lateral flow assay kit | | 3(33.3) | 6(66.7) |
| Review of all final test results by a second person | | 7(77.8) | 2(22.2) |
| Avail UPS for Incubator | | 3(33.3) | 6(66.7) |
| UPS avail for MGT machine | | 9(100) | 0 |
| QC for LPA | | | |
| IQC runs with each batch of specimens/cultures or one batch per week. | | 9(100) | 0 |
| Areas for master-mix preparation, DNA template addition/PCR, and detection are physically separated. | | 9(100) | 0 |
| For the MTBDRplus assay and /or MTBDRsl assay, MTB must be detected for INH and RIF results to be valid. | Amplification control | 9(100) | 0 |
| | Conjugate control | 9(100) | 0 |
| | TUB | 9(100) | 0 |
| | Locus control | 9(100) | 0 |
| For the MTBDR*plus* and /or MTBDR*sl* assays, if invalid results are obtained, repeat with newly extracted DNA from the same specimen. | | 8(88.9) | 1(11.1) |
| Develop LPA TAT | From sediment | 9(100) | 0 |
| | From culture | 9(100) | 0 |
| QC testing new lots of LPA kits with appropriate controls before using. | | 3(33.3) | 6(66.7) |
| Monitoring of expired date and storage condition of kits | | 5(55.6) | 4(44.4) |

*(Continued)*

**Table 3.** (Continued)

| Activities | | Frequency and percent | |
|---|---|---|---|
| | | Yes, n (%) | No, n(%) |
| Routine maintenance and cleaning | Heat block, | 5(55.6) | 4(44.4) |
| | Thermal Cycler | 5(55.6) | 4(44.4) |
| | PCR hood | 5(55.6) | 4(44.4) |
| | TwinCubator | 6(66.7) | 3(33.3) |
| | Water bath | 5(55.6) | 4(44.4) |
| | Microcentrifuge | 5(55.6) | 4(44.4) |
| Calibration of Automatic pipette | | 6(66.7) | 3(33.3) |
| Review of all final test results by a second person | | 6(66.7) | 3(33.3) |
| UPS available for twincubetor | | 6(66.7) | 3(33.3) |

MGIT: Mycobacteria growth indicator tube, BHI: brain-heart Infusion; LJ: Löwenstein–Jensen; PBS: Phosphate-buffered saline: NALC/NaOH: N-acetyl-l cysteine–sodium hydroxide; NTM: Non-tuberculosis mycobacteria; MTB: Mycobacterium Tuberculosis; IQC: Internal quality control; UPS: Uninterruptible power supply; n: Number of the participant; PCR: Polymerase chain reaction; DNA: Deoxyribonucleic acid; MTBDR: Mycobacterium tuberculosis drug resistance; INH: Isoniazid; TAT: Turnaround time; QC: quality control; LPA: Line probe Assay; RIF: Rifampicin; UPS: uninterruptible power supply; IQC: Internal quality control.

**Table 4. External quality assurance practice in selected TB-diagnostic health facilities and TB-culture laboratories.**

| Participation schemes | | Microscopy n (%) | GeneXpertMTB/RIF assay n (%) | Culture and ID n(%) | LPA n (%) |
|---|---|---|---|---|---|
| Number of blind rechecking participant HF | | 18/34(52.9) | N/A | N/A | NA |
| Blind rechecking participation per year | Annually | 3/18(16.75) | N/A | N/A | N/A |
| | Bi-annual | 2/18(11.1) | N/A | N/A | N/A |
| | Quarterly | 13/18(72.2) | N/A | N/A | N/A |
| PT participant HF | | 27/34(79.4) | 34(100) | 8/9(88.9) | 9(100) |
| PT participation per a year | Annually | 3/27(12) | 4/34(11.8) | 4/8(50) | 2/9(22.2) |
| | Bi-annual | 7/27(28) | 19/34(55.9) | 4/8(50) | 4/9(50) |
| | Every fourth month | 9/27(36) | 4/34(11.8) | 0 | 1/9(11.1) |
| | Quarterly | 6/27(24) | 6/34(17.6) | 0 | 0 |
| | Unknown | 2/27(7.4) | 1/34(2.9) | 0 | 2/9(22.2) |
| Onsite supervision practice | | 30/34(88.2%) | | 3/6 (33.3) | |
| **EQA participation challenge among EQA participants of TB diagnostic health facilities** | | | | | |
| Leakage | | NA | 2/34(5.9) | 0 | 0 |
| PT sample volume | | NA | 2/34(5.9) | 1(12.5) | 0 |
| Unclear procedure instruction | | 0 | 0 | 1(12.5) | 1(12.5) |
| Commitment of staff | | 2(7.4) | 4(11.8) | 0 | 0 |
| Interruption of PT sample | | 1(3.7) | 7(20.6) | 0 | 0 |
| Delay in transportation | | 11(40.7) | 4(11.8) | 2(25.0) | 3(33.3) |
| Communication gap with the provider | | 8(29.6) | 10(29.4) | 0 | 1(12.5) |
| Lack of standard request and reporting paper | | 2(7.4) | 4(11.8) | O | 0 |
| Delays in feedback receiving | | 5(18.5) | 12(35.3) | 2(25.0) | 3(33.3) |
| Feedback missing | | 9(33.3) | 13(38.2) | 0 | 0 |
| Delay in reporting | | 6(22.2) | 9(26.5) | 1(12.5) | 3(33.3) |

RIF: Rifampicin; NA: None applicable; MTB: *Mycobacterium Tuberculosis*; LPA: Line probe Assay; ID: Identification: PT: panel testing, n:Number of the participant; HF: Health Facility.

international sample transportation rules and regulations [38]. It also disagrees with the WHO Laboratory Quality Management System handbook that, the laboratory should check sample quality, including its volume, container, and storage condition [39]. The reference laboratory should keep track of the sample collection date for reception and must inspect the specimen transport box to ensure that specimens remain cool during transportation [20].

The study participants' equipment calibration habits were poor in the majority of health facilities. For example, regular thermometry and GeneXpert machine calibration were performed only in 29.4% and 58.8% of Xpert® MTB/RIF testing health facilities respectively. A similar problem occurred in TB-culture laboratories. Only 55.6% of TB-culture laboratories had the experience of regular calibration of negative pressure, while negative pressure is the minimum design feature necessary to safely manipulate TB cultures [40]. Performing regular preventive maintenance and calibration for each piece of equipment is important since it has a direct effect on the laboratory results [20].

This study observed that a lot-to-lot verification and method verification SOP were not developed in most Xpert® MTB/RIF assay and TB-Culture diagnostic laboratories. SOPs for each that testing procedure in the laboratory are used to describe in detail the complete technique for performing tests and ensuring the consistent and reproducible results will be generated [41]. Thus, all TB diagnostic health facilities are expected to have SOPs relating to their services.

In our findings, the number of health facilities that run IQC of AFB smear is much higher (50%) than in a study conducted in Southern Ethiopia, with 12.5% of tuberculosis laboratories [42] and Western Amhara (45%) of tuberculosis laboratories [8]. The results of the outcome may vary because of study site differences. This study included only Xpert® MTB/RIF testing sites from all over the country.

This study also identified a small number of Xpert® MTB/RIF testing health facilities, which were observed monitoring internal controls, probe check control, and specimen processing control. The Xpert® MTB/RIF assay system automatically performs internal quality control (sample processing control and Probe Check Control) for each sample [43]. While the machine is performing internal quality control, laboratory experts should monitor the machine's internal controls, probe check control, and Specimen Processing Control [44].

The use of a combination of both liquid and solid culture media gives the most optimal rates of mycobacterial recovery [17]. Ethiopian TB-culture laboratories use both liquid and solid culture media. During TB-culture processing, 100% of TB-culture laboratories included a negative (start and end) control and 33.3% of TB-Culture laboratories were running a positive control. The TB-culture laboratories should inoculate a negative control in an MGIT tube and LJ tube to check for sterility of the processing reagents and monitor for cross-contamination and a positive sputum processing control (suspended pellet) to monitor NaOH killing of MTB [17].

According to this finding, method verification practices for MGIT machines and GeneXpert machines in Ethiopia's TB diagnostic laboratories were less than 50%, even though adhering to the new testing method is required to know its' functionality in each laboratory setup [17, 20].

Our findings showed that a lot-to-lot of verification practice for GeneXpert® MTB/RIF assay was performed in only 8.8% of Xpert® MTB/RIF assay testing laboratories. Similarly, in TB-culture laboratories, lot-to-lot verification practices were much lower for LPA kits (33.3%). It is recommended that the laboratory personnel should evaluate the new lots of reagents with previous lots in his/her laboratory to reduce variation between different lots with the point of manufacture [20, 45].

In Ethiopia, TB diagnostic health facilities implement the three types of EQA programs (blind rechecking, proficiency testing, and onsite supervision) [18, 36]. This study has shown

that EQA-blind rechecking participation practice for AFB-Microscopy was 52.9% among Xpert® MTB/RIF testing health facilities. This finding is similar to the northern part of Ethiopia's that showed participating decreased from time to time which indicates there should be a re-vitalization of the EQA-blind rechecking practice in the country [46]. Regarding the on-site supervision practice, on-site supervision was not well practiced in TB-culture laboratories (33.3%). Usually, regional laboratories in Ethiopia are supervised by the NTRL, NTP, or partners. Regular on-site supervision helps teams to discuss concerns and solve problems. Therefore, scheduling site visits are important to ensure regular onsite supervision [20, 47].

This assessment identifies several challenges to EQA-PT participation; delays in the transportation of PT samples were a major issue in AFB-PT participation. During the study period, the PT samples were transported by the postal system or by TB focal person.

The communication gap with the provider is lower in TB culture laboratories compared to the GeneXpert® MTB/RIF assay. From the author's view, this may be due to the presence of few TB-Culture laboratories and frequent contact with the provider for other work in addition to these activities. However, communication has not resolved the issue of missing EQA-feedback and delays in reporting to the provider, which requires further investigation and the establishment of a system for result delivery and reporting.

## Limitations of the study

This study did not show the consistency or regular practice of quality assurance activities in health facilities. Most of the findings are not compared with other similar studies because of limited studies in the area. Rather, the findings are discussed with national and international guidelines that recommended to have a quality-assured laboratory service. Additionally, the actual results of quality assurance practices or scheme participation and challenge in implementing IQC, a lot-to-lot verification, method verification, and equipment maintenance were not observed.

## Conclusion

Based on the assessment findings, TB culture and identification, and LPA were performed in most TB-culture laboratories, while IQC for AFB microscopy, lot-to-lot verification, method validation, and equipment calibration were not practiced. Most Ethiopian TB diagnostic health facilities participate in the EQA-PT. However, there were difficulties during PT participation, including PT-transportation delays, communication gaps with the provider, delays in feedback from the provider, missing feedback, and reporting to the provider.

Based on the finding, the authors strongly recommend performing internal quality control in all AFB smear microscopy testing health facilities, also performing a lot-to-lot verification for purchased reagents and batch to batch for in-house reagents. Additionally, method validation, regular machine calibration, and maintenance need critical consideration to be practiced. Furthermore, creating a system to avoid delays in feedback missing, result receiving, and delivery will help solve the problem. Within this, the authors also recommend further studies to be performed to identify the challenge of practicing quality assurance activities in TB diagnostic health facilities in depth.

## Acknowledgments

We would like to acknowledge the contribution of all the participant health facilities for their cooperation during the data collection process. We would like to extend our appreciation to all national TB research teams for their participation in the data collection.

## Author Contributions

**Conceptualization:** Yeshiwork Abebaw.

**Data curation:** Yeshiwork Abebaw, Abebaw Kebede, Kirubel Eshetu, Ephrem Tesfaye, Mengistu Tadesse, Waganeh Sinshaw, Misiker Amare, Dinka Fikadu Gamtesa, Betselot Zerihun, Melak Getu, Getachew Seid, Anteneh Yalew, Getu Diriba.

**Formal analysis:** Yeshiwork Abebaw, Misiker Amare, Dinka Fikadu Gamtesa, Anteneh Yalew.

**Funding acquisition:** Ephrem Tesfaye.

**Investigation:** Yeshiwork Abebaw, Abebaw Kebede, Kirubel Eshetu, Ephrem Tesfaye, Mengistu Tadesse, Waganeh Sinshaw, Misiker Amare, Dinka Fikadu Gamtesa, Betselot Zerihun, Melak Getu, Getachew Seid, Anteneh Yalew, Getu Diriba.

**Methodology:** Yeshiwork Abebaw, Abebaw Kebede, Kirubel Eshetu, Mengistu Tadesse, Waganeh Sinshaw, Misiker Amare, Dinka Fikadu Gamtesa, Anteneh Yalew.

**Software:** Anteneh Yalew.

**Validation:** Yeshiwork Abebaw, Abebaw Kebede, Kirubel Eshetu, Dinka Fikadu Gamtesa, Anteneh Yalew.

**Writing – original draft:** Yeshiwork Abebaw.

**Writing – review & editing:** Yeshiwork Abebaw, Abebaw Kebede, Kirubel Eshetu, Anteneh Yalew, Getu Diriba.

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
