## [Decision Letter · Decision Letter 0]

2 Dec 2021

PONE-D-21-20924Quality assurance practices in tuberculosis diagnostic health facilities in EthiopiaPLOS ONE

Dear Dr. Abebaw,

Thank you for submitting your manuscript to PLOS ONE. After careful consideration, we feel that it has merit but does not fully meet PLOS ONE’s publication criteria as it currently stands. Therefore, we invite you to submit a revised version of the manuscript that addresses the points raised during the review process.

Apart from all the comments that the reviewers have already pointed out, I want to emphasize a few of them and make a few comments as well:

Line 170-171- 'Moreover, almost 50% of TB-culture laboratories, 4/9 (44.4%) were no regular negative pressure calibration'. In this sentence, 'were' should be replaced by 'did not have'

Reviewer 1 has already pointed out that there is no Figure 2 which is mentioned in text whereas Table 2 is missing from the main text.

Line 232, 233- ISO Number to be corrected

Line 241- 'collation' should be 'collection'

Line 244- 'pre-pre and pre-analytical' should be corrected.

Table 1- SLIMTA to be replaced by SLMTA

We look forward to receiving your revised manuscript.

Kind regards,

Shampa Anupurba, MD

Academic Editor

PLOS ONE

https://journals.plos.org/plosone/s/file?id=ba62/PLOSOne_formatting_sample_title_authors_affiliations.pdf"

2. Please provide additional details regarding participant consent. In the ethics statement in the Methods and online submission information, please ensure that you have specified whether consent was written or verbal/oral. If consent was verbal/oral, please specify: 1) whether the ethics committee approved the verbal/oral consent procedure, 2) why written consent could not be obtained, and 3) how verbal/oral consent was recorded. If your study included minors, please state whether you obtained consent from parents or guardians in these cases. If the need for consent was waived by the ethics committee, please include this information.

3. Please include in your Methods section (or in Supplementary Information files) the participating hospitals/institutions.

4. Please include additional information regarding the survey or questionnaire used in the study and ensure that you have provided sufficient details that others could replicate the analyses. For instance, if you developed a questionnaire as part of this study and it is not under a copyright more restrictive than CC-BY, please include a copy, in both the original language and English, as Supporting Information.

5.  We suggest you thoroughly copyedit your manuscript for language usage, spelling, and grammar. If you do not know anyone who can help you do this, you may wish to consider employing a professional scientific editing service.

Reviewers' comments:

Reviewer's Responses to Questions

**Comments to the Author**

1. Is the manuscript technically sound, and do the data support the conclusions?

Reviewer #1: Yes

Reviewer #2: Partly

Reviewer #3: No

2. Has the statistical analysis been performed appropriately and rigorously? 

Reviewer #1: Yes

Reviewer #2: Yes

Reviewer #3: No

3. Have the authors made all data underlying the findings in their manuscript fully available?

Reviewer #1: Yes

Reviewer #2: Yes

Reviewer #3: No

4. Is the manuscript presented in an intelligible fashion and written in standard English?

Reviewer #1: No

Reviewer #2: Yes

Reviewer #3: No

5. Review Comments to the Author

Reviewer #1: Dear Authors, Your study is very valuable for reliability of TB laboratories' results. You need some corrections in this article. I have also some suggestions for you. I showed corrections and suggestions some areas on the text. First of all, you should check some sentences in terms of writing language.For example; In 154. line; Among the selected laboratories, 27/34 (79.4%) of GeneXpert®.....You can say 27 (79,4%)Gene Xpert Lab.....

There are 2 kinds of Figure 1, but no figure 2 (see 490. and 495. lines!). You should write "Figure" do not abbreviate, like Fig! It is not required Method verification for FM or ZN. But you should check QC for staining solutions, slide preparation and evaluation. So for all, there are negative- and positive- slide controls known results! Technician should be run positive and negative slide control in every work, and record the their results. The result is okay, then should be continue patient slides. Your sample size for TB labs is written 34 +9 (43). But in 151. line, it is written totally 40 labs. You say 31 labs for TB-culture... In 180. line, it refers to Table 3. Table 2 should be placed in text prior to Table 3. There is no Table 2 in text, except separete a Table 2! In 189. line, ...a lot lot verification....please correct this! In 217. line; On-site supervision practice was not well practiced in TB-culture laboratories, only 3/6 (33.3)....As sample size, it is written 9 TB- culture labs! But here is 6 labs! In 288. line, negative (star and end ) control....please corrected star as start. In 509. line, correct avail asr available. In 490. line, Figure 1; lot to lots and annua.....please correct the. In 365. line; references; Please check "reference writing rules" from guidelines for authors; Is journal writing stile italic or not!

Reviewer #2: Please see attached.

Reviewer #3: Thank you editor sir for giving me the opportunity to review the article.

I have gone through the same and need replay for the below details:

1. Authors mentioned the numbers of participants lab @ 43 at some time and 34, 40 at others. So they can clarify it

2. Is it ISO 18519 or ISO 15189? Authors mentioned both

3. Recommendation on line 297-298 are ambiguous. It is very unclear

4. Authors mentioned checklist on line133. I would like to have a look at this

5. Paragraph@ lines 303-307 should in line with the standard format of "discussion". Here, it is uncertain about final messages

6. Lines @ 320-323 sound contrary to lines @ 50-52

6. PLOS authors have the option to publish the peer review history of their article (what does this mean?). If published, this will include your full peer review and any attached files.

Reviewer #1: **Yes: **Hülya Şimşek

Reviewer #2: No

Reviewer #3: No

---

## [Author Response · Author response to Decision Letter 0]

18 Jan 2022

Dear Editors and reviewers 

Thank you for your informative comments and thoughtful comments on our manuscript. We have addressed your and the reviewers comments one by one. We also appreciate the reviewers for their critical observation and informative comments which radically improved our manuscript quality. We have provided point-by-point response to the reviewers' comments below with table. Also, we would like to inform you that we have used hyperlink to indicate where we made changes in the previous version of the manuscript based on the reviewers’ comments.

---

## [Decision Letter · Decision Letter 1]

16 Mar 2022

PONE-D-21-20924R1Quality assurance practices in tuberculosis diagnostic health facilities in EthiopiaPLOS ONE

Dear Dr. Abebaw,

Thank you for submitting your manuscript to PLOS ONE. After careful consideration, we feel that it has merit but does not fully meet PLOS ONE’s publication criteria as it currently stands. Therefore, we invite you to submit a revised version of the manuscript that addresses the points raised during the review process. Specifically, reviewers raised concerns on the English language usage and grammar. Please note that PLOS ONE does not provide copyediting or proofs of accepted manuscripts. We therefore recommend that you carefully review your manuscript and correct any errors at this time. We will carefully review your manuscript upon resubmission, so please ensure that your revision is thorough.

We look forward to receiving your revised manuscript.

Kind regards,

Jianhong Zhou

Associate Editor

PLOS ONE

Journal Requirements:

Reviewers' comments:

Reviewer's Responses to Questions

**Comments to the Author**

1. If the authors have adequately addressed your comments raised in a previous round of review and you feel that this manuscript is now acceptable for publication, you may indicate that here to bypass the “Comments to the Author” section, enter your conflict of interest statement in the “Confidential to Editor” section, and submit your "Accept" recommendation.

Reviewer #1: All comments have been addressed

Reviewer #2: All comments have been addressed

2. Is the manuscript technically sound, and do the data support the conclusions?

Reviewer #1: Yes

Reviewer #2: Yes

3. Has the statistical analysis been performed appropriately and rigorously? 

Reviewer #1: Yes

Reviewer #2: Yes

4. Have the authors made all data underlying the findings in their manuscript fully available?

Reviewer #1: Yes

Reviewer #2: Yes

5. Is the manuscript presented in an intelligible fashion and written in standard English?

Reviewer #1: Yes

Reviewer #2: No

6. Review Comments to the Author

Reviewer #1: (No Response)

Reviewer #2: Thank you for your point-by-point responses to the reviewers’ comments which I believe have addressed all the issues raised. The following need to be corrected:

Line 95. “blinded rechecking of TB diagnostic centers”. Is this blinded rechecking of TB slides?

Line 174. “Date” should be replace by “data”.

In addition, there is need for minor edits and in general the text needs to be streamlined so the language can flow. I have given a few examples below, but there are several more instances in the manuscript.

Line 75. Should read: “…quality assurance of testing…”

Lines 79-80. This sentence remains unclear. Suggestions in italics: “Quality control ensures that all operations in the TB diagnostic laboratory are monitored, including equipment checks and checking new lots of staining solutions

Line 81. “EQA plays a vital role in monitoring laboratory performance…”

Line 82. ”… quality services.”

Lines 83-84. “One of the End TB strategic goals in 2020 was to enroll all TB diagnostic facilities in EQA”.

Line 89. “…measurable laboratory improvement in a resource limited settings”.

Line 91. “Ethiopia National TB prevention and control program with…”. If this is the title of a program, the relevant words should start with capital letters.

7. PLOS authors have the option to publish the peer review history of their article (what does this mean?). If published, this will include your full peer review and any attached files.

Reviewer #1: No

Reviewer #2: No

---

## [Author Response · Author response to Decision Letter 1]

15 Apr 2022

Dear Editor and reviewers,

Thank you for your informative comments and thoughtful comments on our manuscript. We have addressed your and the reviewer comments one by one. We also appreciate the reviewers for their critical observation and informative comments which radically improved our manuscript quality. We have provided a point-by-point response to the reviewer 2 comments below with a table. Also, we would like to inform you that we have used hyperlink to indicate where we made changes in the previous version of the manuscript based on the reviewers’ comments.

---

## [Editor Report · Decision Letter 2]

25 May 2022

Quality assurance practices in tuberculosis diagnostic health facilities in Ethiopia

PONE-D-21-20924R2

Dear Dr. Abebaw,

We’re pleased to inform you that your manuscript has been judged scientifically suitable for publication and will be formally accepted for publication once it meets all outstanding technical requirements.

Kind regards,

Sarman Singh, MD, FRSC, FRCP

Academic Editor

PLOS ONE

Additional Editor Comments (optional):

None
---

## [Editor Report · Acceptance letter]

1 Jun 2022

PONE-D-21-20924R2 

Quality assurance practices in tuberculosis diagnostic health facilities in Ethiopia 

Dear Dr. Abebaw:

I'm pleased to inform you that your manuscript has been deemed suitable for publication in PLOS ONE. Congratulations! Your manuscript is now with our production department. 

Kind regards, 

on behalf of

Professor Sarman Singh 

Academic Editor

PLOS ONE